# Long-Term Effects of Microfiltered Seawater and Resistance Training with Elastic Bands on Hepatic Parameters, Inflammation, Oxidative Stress, and Blood Pressure of Older Women: A 32-Week, Double-Blinded, Randomized, Placebo-Controlled Trial

**DOI:** 10.3390/healthcare12020204

**Published:** 2024-01-15

**Authors:** Carlos Babiloni-Lopez, Pedro Gargallo, Alvaro Juesas, Javier Gene-Morales, Angel Saez-Berlanga, Pablo Jiménez-Martínez, Jose Casaña, Josep C. Benitez-Martinez, Guillermo T. Sáez, Julio Fernández-Garrido, Carlos Alix-Fages, Juan C. Colado

**Affiliations:** 1Research Group in Prevention and Health in Exercise and Sport (PHES), Department of Physical Education and Sports, University of Valencia, 46010 Valencia, Spain; carbalo3@alumni.uv.es (C.B.-L.); pedro11gb@gmail.com (P.G.); alvaro.juesastorres@uchceu.es (A.J.); angel.saez@uv.es (A.S.-B.); pablowlfit@gmail.com (P.J.-M.); guillermo.saez@uv.es (G.T.S.); carlosalixentrenamientos@gmail.com (C.A.-F.); juan.colado@uv.es (J.C.C.); 2ICEN Institute, 28840 Madrid, Spain; 3Exercise Intervention for Health Research Group (EXINH-RG), University of Valencia, 46010 Valencia, Spain; jose.casana@uv.es; 4Research Group in Physiotherapy Technology and Recovering (FTR), University of Valencia, 46010 Valencia, Spain; josep.benitez@uv.es; 5Service of Clinical Analysis, University Hospital Dr. Peset—FISABIO, 46017 Valencia, Spain; 6Department of Biochemistry and Molecular Biology, Faculty of Medicine and Odontology, University of Valencia, 46010 Valencia, Spain; 7Nursing Department, Faculty of Nursing and Chiropody, University of Valencia, 46010 Valencia, Spain; julio.fernandez@uv.es; 8Applied Biomechanics and Sport Technology Research Group, Department of Physical Education, Autonomous University of Madrid, 28049 Madrid, Spain

**Keywords:** ageing, female, postmenopausal, resistance training, elastic bands, mineral-enriched supplement, liver, oxidative stress, inflammatory status, hypertension

## Abstract

The bulk of research on microfiltered seawater (SW) is based on its short-term effects. However, the long-term physiological adaptations to combining SW and resistance training (RT) are unknown. This study aimed to analyse the impact of an RT program using elastic bands combined with SW intake on hepatic biomarkers, inflammation, oxidative stress, and blood pressure in post-menopausal women. Ninety-three women voluntarily participated (age: 70 ± 6.26 years; body mass index: 22.05 ± 3.20 kg/m^2^; Up-and-Go Test: 6.66 ± 1.01 s). RT consisted of six exercises (32 weeks, 2 days/week). Nonsignificant differences were reported for hepatic biomarkers except for a reduction in glutamic-pyruvic transaminase (GPT) in both RT groups (RT + SW: *p* = 0.003, ES = 0.51; RT + Placebo: *p* = 0.012, ES = 0.36). Concerning oxidative stress, vitamin D increased significantly in RT + SW (*p* = 0.008, ES = 0.25). Regarding inflammation, interleukin 6 significantly decreased (*p* = 0.003, ES = 0.69) in RT + SW. Finally, systolic blood pressure significantly decreased in both RT groups (RT + placebo: *p* < 0.001, ES = 0.79; RT + SW: *p* < 0.001, ES = 0.71) as did diastolic blood pressure in both SW groups (RT + SW: *p* = 0.002, ES = 0.51; CON + SW: *p* = 0.028, ES = 0.50). Therefore, RT + SW or SW alone are safe strategies in the long term with no influences on hepatic and oxidative stress biomarkers. Additionally, SW in combination with RT positively influences vitamin D levels, inflammation, and blood pressure in older women.

## 1. Introduction

Specific nutritional supplementation and resistance training (RT) have shown positive effects as adjuvant methodologies in the prevention and treatment of numerous chronic conditions [1,2]. Nutritional supplements combined with appropriate dietary intake can enhance sports performance, health status, and well-being [3,4]. In particular, microfiltered seawater (SW) is a mineral-enriched supplement that could be effective in preventing and treating several health conditions such as metabolic syndrome [5]. Human research on the potential therapeutic effects of SW has emerged in the last few years [6,7]. However, due to the high levels of sodium that SW contains, its beneficial impact on older adults should be cautiously interpreted [8]. Senescent changes lead to an increase in hypertension due to vascular resistance, which can impair the pump activity of the sodium membrane [9]. Therefore, high-sodium diets may increase blood pressure and risk for liver and kidney diseases [10]. Considering that no long-term studies about the benefits or risks of SW on hepatic biomarkers have been conducted in older adults, novel research approaching this topic may be valuable.

RT with elastic bands (EBs) entails positive short- and long-term adaptations in individuals with diverse characteristics, including postmenopausal women [11,12,13,14]. One relevant aspect of RT programs is that strenuous exercise increases energy expenditure and oxygen consumption [15]. Hence, during strenuous exercise, practitioners enter a hypoxic state, which boosts reactive oxygen species (ROS) production [15]. Consequently, the natural balance between antioxidant and oxidative systems is disturbed, which can alter performance due to reactive radicals’ aggregation [16]. Besides, the relative reduction of antioxidant enzymes and the agglomeration of oxidative metabolites generate lipid peroxidation, which finally leads to oxidative damage [16]. In addition, free radical damage may result in musculoskeletal dysfunctions in individuals with magnesium (Mg) [17] and vitamin D [18] deficits.

For instance, the absence of Mg is linked to cytokine and ROS production, increasing the exercise-induced inflammatory response [19]. Likewise, interleukin 6 (IL-6) is involved in the pathogenesis and clinical evolution of different diseases such as atherosclerotic vascular disease, arthritis, osteoporosis, sarcopenia, or dynapenia [20]. On the other hand, vitamin D is also a potent antioxidant that allows for balanced mitochondrial activity and prevents DNA damage and lipid peroxidation [21]. A deficit of vitamin D (hypovitaminosis) augments the severity or incidence of certain conditions associated with age, e.g., impaired response to insulin, glycemic dysregulation, increased blood pressure, obesity, osteoporosis, and systemic inflammatory diseases [21]. Collectively, SW may reduce oxidative stress [17]. However, to date, only rats with Mg deficit have been exposed to SW supplementation in the long term [17]. Furthermore, despite the important role of minerals in oxidative stress, little is known about the SW and RT relationship in older adults.

With this in mind, the aim was to explore the influence of supplementation with microfiltered SW and RT with EBs on hepatic biomarkers (glutamic-oxaloacetic transaminase [GOT], glutamic-pyruvic transaminase [GPT], gamma-glutamyl transpeptidase [GGT], alkaline phosphatase enzyme [ALP]), oxidative stress (malondialdehyde [MDA], oxidized/reduced glutathione ratio [GSH/GSSG]) and related parameters (vitamin D, 25OHD type), inflammatory status (interleukin 6 [IL-6]), and systolic (SBP) and diastolic blood pressure (DBP) in older women. It was hypothesized that (i) the intake of SW added to the RT program with EBs would not alter hepatic biomarkers [22], (ii) oxidative stress would be ameliorated in those participants who ingest SW [17], (iii) vitamin D would be increased with SW intake [23], (iv) IL-6 would be reduced with SW supplementation + RT, and (v) blood pressure would not significantly differ between both intervention groups [24].

## 2. Materials and Methods

### 2.1. Experimental Design

A 32-week randomized controlled trial was applied following the Proper Reporting of Evidence in Sport and Exercise Nutrition Trials (PRESENT) (Appendix A). Measurements were conducted pre- and post-intervention. Subjects were randomly allocated to groups, and concealment was applied for the allocation of subjects, procedures, and analyses. This study is part of a larger research project authorized by the Institutional Review Board of the University of Valencia to explore the influences of diverse training loads on health, functionality, and quality of life. Two studies derived from this project have already been published [16,25] and more studies are being conducted based on this project [26]. The study was conducted according to the tenets of the Declaration of Helsinki. The volunteers included in the study gave informed consent and were allowed to abandon the research at any time.

Participants were encouraged to maintain their usual life habits during the study. This was ensured by oral communication with each participant during the training sessions and visits to the facilities. The participants were divided into 4 groups (RT + SW, RT + placebo [PLA], control [CON] + SW, and CON + PLA). External staff associated an alphanumeric code with each participant and training group to blind the researchers. To obtain blood samples and blood pressure measurements, participants attended the laboratories on two different days separated by 48 h.

### 2.2. Participants

We determined the minimum number of participants with the software G* Power 3.1 [27], using the commands: F-tests/ANOVA: repeated measures/mixed design. We obtained an optimal sample size of 72 participants for a power (1–ß) of 0.80, α = 0.05, and effect size (ES) of 0.35. The ES was calculated based on the outcomes of previous studies from our research group.

The participants were enrolled through an open advertisement placed in different activity centers for older people in Valencia (Spain). The eligibility criteria included (i) demographical criteria: (a) females equal or above 65 years; (ii) physical criteria: (b) being able to climb 10 stairs without stopping, (c) walking 100 m without a walker, and (d) practicing less than 1 h of physical activity per week within the six months before the start of the study; (iii) psychological criteria: (e) obtaining at least 23 points in the mini-mental state examination; (iv) medical criteria: participants were excluded if: (f) they presented any cardiovascular, neurological, renal, metabolic, hepatic, musculoskeletal or pulmonary disorders and/or (g) they were treated with any type of drug or supplement (e.g., beta-blockers, estrogens, vitamins, etc.) that could bias the outcomes of the study.

The initial sample was composed of 160 Caucasian women. However, 51 participants were excluded for the following reasons: (i) 19 refused to participate due to lack of interest and (ii) 32 did not fulfil the eligibility criteria—(a) 5 presented specific diseases, (b) 11 were following medications that could bias the results, (c) 2 presented ages under the inclusion criteria, (d) 1 planned to leave the area during the intervention, (e) 6 refused to participate due to schedule incompatibilities, (f) 2 were already engaged in resistance training programs, and (g) 1 presented a result in the mini-mental state examination lower than the cutoff score. Therefore, 109 participants were randomized employing random digital allocation by an independent researcher not involved in the study.

### 2.3. Training Intervention

Two familiarization sessions were conducted to obtain anthropometric measurements, learn exercises [28], and establish the grip width of the elastic band for each subject and exercise (see [29]). Volunteers were asked to perform sets of 6 and 15 repetitions with an EB (Theraband, Hygenic Corporation, Akron, OH, USA; five colors [increasing in resistance]: green, blue, black, silver, and gold) at different grip widths (i.e., different rate of perceived exertion [RPE] values).

Participants completed two weekly sessions of 55–60 min on non-consecutive days (separated by 48–72 h) for 32 weeks. The control group was not involved in any exercise programs. The loads were modified (adapting the color of the band and width of the grip) to maintain the appropriate training intensity. The intensities used were: (i) high (6 repetitions equivalent to 85% of one-repetition maximum [1RM]) and (ii) moderate (15 repetitions equivalent to 65–70% 1RM). Within the sets, a progression in the OMNI-RES EB intensity was established from 6–7 scores (“somewhat hard”) in the first four weeks to 8–9 scores (“hard”) in the following 28 weeks of the intervention [29]. We controlled the speed of execution with a metronome throughout the whole session. The execution tempo consisted of a two-second concentric phase and a two-second eccentric phase. More detailed information about the specific procedures can be found in Juesas et al. [25].

### 2.4. Supplementation Protocol

The supplement and placebo were indistinguishable in their organoleptic properties. The supplementation protocol consisted of the ingestion of 20 mL of the supplement or placebo before each training session. The mineral-enriched supplement was composed of 11.87 g L^−1^ sodium; 20.36 g L^−1^ chloride; 1.36 g L^−1^ magnesium; 433 mg L^−1^ calcium; 441 mg L^−1^ potassium; 148 mg L^−1^ bicarbonate; 11.8 μg L^−1^ zinc; 116.9 μg L^−1^ manganese; and 6.6 μg L^−1^ cupper. Moreover, other chemical components of seawater were present in the supplement in trace magnitudes: proteins, lipids, and water- and fat-soluble vitamins (D-biotin, thiamine, riboflavin, nicotinamide, cyanocobalamin, pyridoxine, retinal, D3, α-tocopherol, and K1). Contraindications were not identified in the use of this product. The placebo drink was exclusively composed of water. The placebo samples were distributed with the same appearance by a blinded researcher. All of the groups received the same feedback regarding the supplement, indicating that they were going to have an enriched mineral water supplement. Both substances were prepared by Laboratories Quinton International S.L. (Alicante, Spain).

### 2.5. Biochemical Blood Parameters Measurement

Participants attended the laboratories between 8:00 and 10:00 am after fasting for 12 h. Venous blood samples (10–15 mL) were collected into EDTA tubes. Samples were refrigerated at 2–4 °C until processing, which was conducted within 4 h. Density gradient centrifugation with Histopaque (Sigma H-1077, Sigma-Aldrich Co. LLC, St. Louis, Missouri, United States) for 30 min at 1700 rpm and 12 °C separated the blood samples into plasma and peripheral blood mononuclear cells (PBMc). These samples were stored at −80 °C. The same procedure was used to obtain the plasma layer stored at −80 °C. Serum separation involved drawing 10 mL whole blood samples into dry tubes with a gel separator and coagulation activator. After clot retraction (15–30 min at room temperature), centrifugation for 5 min at 3500 rpm and 4 °C yielded serum supernatant. The supernatant was aliquoted and frozen at −80 °C. The process was performed in duplicate, with measurements differing by over 15% being repeated. Data analysis used the average of both readings.

Specifically for MDA, lipoperoxides and thiobarbituric acid adduction products were processed via boiling in diluted orthophosphoric acid. The resulting compounds were quantified using spectrophotometry. For MDA quantification, PBMc samples were mixed with H_3_PO_4_ and thiobarbituric acid solution, heated, neutralized, and centrifuged. MDA was purified and measured using HPLC-UV with an isocratic elution method. Chromatographic assays were performed for 3 min at a flow rate of 1 mL/min. Compounds were quantified at 532 nm. Total protein content was determined using the Lowry method. Outcomes were measured in nmol/mg. The coefficient of variation was 3.21% with a 95% confidence interval between 1.9% and 4.51%. Finally, the thiol state (GSH/GSSG) was assessed through spectrophotometry according to the Cayman Chemical (Ann Arbor, MI, USA) assay kit procedures: GSH and GSSH (no. 703002).

The minimal clinically important difference (MCID) (i.e., the smallest variation required to produce clinically relevant results, see Section 2.7) of hepatic variables was 1.12 for GOT, 1.57 for GPT, 2.55 for GGT, and 3.38 for ALP. For the oxidative stress biomarkers, it was 0.01 for MDA and 0.09 for GSH/GSSG. The MCID for the inflammatory biomarker IL-6 was 0.34, and for Vitamin 25OHD it was 2.40.

### 2.6. Blood Pressure Measurement

Assessments of resting SBP and DBP were conducted at the same time of day (in the morning). Participants remained seated for 15 min, with arms relaxed at the sides of the body, in a quiet atmosphere between 24 and 26 °C. SBP and DBP were obtained with a digital sphygmomanometer (OMRON M3, OMRON Corporation, Kyoto, Japan).

MCID values of 3.08 and 1.70 mmHg were obtained for the SBP and DBP, respectively.

### 2.7. Statistical Analysis

The intention-to-treat method was used in all the analyses of the current study. All values are presented as the mean and standard deviation (SD). A uniform cut-off criterion of *p* < 0.05 was set as statistically significant. Statistical analyses were conducted using the software SPSS (IBM SPSS version 25.0, Chicago, IL, USA).

All variables followed a normal distribution when the Kolmogorov–Smirnov test was used. Therefore, a two-way mixed analysis of variance (ANOVA) of repeated measures analyzed the effects of the group (RT + SW, RT + PLA, CON + SW, CON + PLA) and time (pre-intervention and post-intervention) on each dependent outcome. In addition, the ES was calculated through eta partial squared (ηp^2^) according to the following interpretation: (a) 0.01 < ηp^2^ < 0.06 small effect, 0.06 ≤ ηp^2^ ≤ 0.14 medium effect, and ηp^2^ > 0.14 large effect. Post-hoc analyses were performed using the Bonferroni adjustment. The ES of the post-hoc comparisons was calculated by means of Cohen’s d, with trivial (<0.20), small (0.20–0.49), moderate (0.50–0.79), or large effects (≥0.80).

Finally, the MCID of the studied variables was difficult to find for postmenopausal women. Therefore, we calculated the MCID by multiplying the pooled baseline SD by 0.20 (corresponding to the smallest effect size) [30].

## 3. Results

### 3.1. Participants

The participants were 93 females with no experience in training (70.00 ± 6.26 years; body mass index: 22.05 ± 3.20 kg/m^2^; up-and-go test: 6.66 ± 1.01 s) who were randomly distributed into four groups: (i) RT + SW (*n* = 35); (ii) RT + PLA (*n* = 35); (iii) CON + SW (*n* = 11); (iv) CON + PLA (*n* = 12). Initially, 109 women were randomly allocated to the study interventions. Of these, 93 started the study. Finally, 77 participants (RT + PLA: 27; RT + SW: 28; CON + PLA: 11; CON + SW: 11) completed the study (dropout rate of 17.2%). The attendance rate was approximately 75%. Figure 1 presents the participant flow through the study and Table 1 their baseline characteristics. There were no between-group differences (*p* > 0.05, ES < 0.06) at baseline in any of the variables.

### 3.2. Hepatic Biomarkers

Table 2 shows the hepatic parameters analyzed in the present study. The ANOVA showed no significant effects of the interaction group×time on the hepatic biomarkers (aspartate aminotransferase [AST/GOT]: *p* = 0.71; alanine aminotransferase [ALT/GPT]: *p* = 0.28; gamma-glutamyltransferase [GGT]: *p* = 0.72; and alkaline phosphatase [ALP]: *p* = 0.70). It is to be noted that both RT groups had their GPT improved (RT + PLA: d = 0.36; RT + SW: d = 0.51), with larger ES for RT + SW. The rest of the variables did not improve in any of the groups. No significant between-group differences were found in any of the hepatic parameters.

### 3.3. Oxidative Stress, Inflammatory State, and Related Parameters (Vitamin D Status)

Table 3 displays the MDA, GSH/GSSG, IL-6, and vitamin 25OHD before and after the intervention. The ANOVA for group×time only showed a significant effect on IL-6 (IL-6: F_(3,86)_ = 3.66, *p* = 0.01; η_p_^2^ = 0.11). The oxidative stress biomarkers and 25OHD vitamin D did not show statistical differences when the interaction group×time was carried out (MDA: *p* = 0.65; GSH/GSSG: *p* = 0.88; vitamin D: *p* = 0.48). Significant differences between CON + SW and the rest of the groups were encountered in the MDA. However, none of the groups significantly modified their MDA or GSH/GSSG. A significant improvement was only seen in the vitamin D (d = 0.25) and IL-6 (d = 0.69) levels of RT + SW.

### 3.4. Systolic and Diastolic Blood Pressure

Table 4 depicts changes in SBP and DBP. There was a significant effect from the interaction group×time on SBP (F_(3,88)_ = 4.11, *p* = 0.009, η_p_^2^ = 0.12). However, this interaction did not influence DBP (*p* = 0.073). Regarding the post-hoc analyses, both RT + PLA (d = 0.79) and RT + SW (d = 0.71) significantly reduced their SBP, with significant differences in the post-intervention values compared to CON + PLA. Only RT + SW (d = 0.51) and CON + SW (d = 0.50) significantly lowered their DBP. RT + SW showed significant differences compared to CON + PLA.

### 3.5. Individual Responses

Figure 2 and Figure 3 present the individual responses to the study variables significantly affected by the interaction group×time (i.e., IL-6, SBP). At a group level, we can see that while RT + SW and CON + SW improved (i.e., decreased) their IL-6 values, RT + PLA and CON + PLA had these values increased (Figure 2). However, interindividual results showed that a considerably large number of participants (20 out of 35) did not present changes or decrease the IL-6 in RT + PLA. Figure 3 shows that the results in terms of SBP are robust, with only 4 and 5 participants having their SBP increased in RT + PLA and RT + SW, respectively. Participants from CON + PLA and CON + SW presented nonhomogeneous results.

### 3.6. Adverse Events

Neither the SW nutritional supplement nor the RT program produced severe or moderate adverse events. Muscle soreness during the initial four weeks in the RT groups was reported by 23 subjects and recorded as a weak adverse event.

## 4. Discussion

To the best of our knowledge, this is the first study to demonstrate that SW intake added to RT or alone has no adverse effects on hepatic and inflammation parameters in the long term. This study aimed to analyze the influence of supplementation with a mineral-enriched supplement and an RT program with EBs on hepatic biomarkers (GOT, GPT, GGT, and ALP), oxidative stress (MDA, GSH/GSSG), inflammation (IL-6), vitamin D status, and blood pressure (SBP and DBP). The main finding was that hepatic parameters, as it was hypothesized, did not exhibit differences (see Table 2). Regarding the second aim, the consumption of a mineral-enriched supplement (i.e., SW) did not improve oxidative stress biomarkers (see Table 3). Therefore, our second hypothesis was rejected. By contrast, vitamin D was significantly increased in RT + SW, confirming the third hypothesis. Additionally, we found a reduction in the inflammatory profile in RT + SW, confirming our fourth hypothesis. Finally, SBP values were ameliorated in both RT groups (RT + PLA and RT + SW), and DBP was only influenced by the consumption of SW (RT + SW and CON + SW) (see Table 4).

### 4.1. Hepatic Responses

AST/GOT, GGT, and ALP values were not significantly modified in any of the study groups (*p* > 0.05). On the other hand, SW ingestion added to RT (RT + SW: *p* = 0.003; ES = 0.51), and only RT (RT + PLA: *p* = 0.012; ES = 0.36) significantly reduced ALT/GPT. Interestingly, CON + SW did not reach significant differences over baseline values in hepatic biomarkers. These findings are partially aligned with preceding research [31] in which the effects of oral magnesium chloride supplementation on liver inflammation outcomes in non-hypertensive obese women (age range: 30–65 years) were assessed. In this study, participants with or without hypomagnesemia ingested a solution of 50 mL of 5% MgCl_2_. After the intervention, the magnesium supplementation group elicited a significant reduction in ALT/GPT levels compared to the control group. Oppositely, we cannot associate the decreases in ALT/GPT with the ingestion of the mineral-enriched supplement.

Concerning the effects of RT on liver biomarkers, a previous study [32] evaluated the impact of RT with EB combined with yoga exercises on ALT/GPT and AST/GOT levels in postmenopausal women. Contrary to our RT + SW group results, Daneshyar et al. [32] did not find differences in ALT/GPT and AST/GOT values in postmenopausal women. However, their training period lasted eight weeks. As both enzymes are delayed indicators of muscle damage [33], a possible explanation for these differences may be linked to the greater volume and duration of the training in our study. Therefore, our intervention would present greater muscular demands.

### 4.2. Oxidative Stress, Inflammatory Responses, and Related Parameters

The RT + SW group significantly improved IL-6 (*p* = 0.003; ES = −0.69) and vitamin D (*p* = 0.008; ES = 0.25) values, while MDA and GSH/GSSG did not reach a significant effect (*p* 0.061 and *p* = 0.209, respectively). None of the variables were significantly altered in RT + PLA (IL-6, *p* = 0.312; MDA, *p* = 0.683; and GSH/GSSG, *p* = 0.732). Similarly, CON + SW did not present changes in IL-6 (*p* = 0.929) and vitamin D (*p* = 0.261). According to the interindividual responses analysis, it should be borne in mind that 20 (out of 35) participants from RT + PLA and 9 (out of 12) participants from CON + SW improved or did not change their IL-6 levels. Collectively, these results may be partially explained by the composition of the SW supplement (1.36 gL^−1^ Mg^2+^, 433 mgL^−1^ Ca^2+^, and 441 mgL^−1^ K), which may have counteracted the reactive species derived from exercise practice [17]. Due to the lack of reference values for oxidative stress biomarkers, comparative assessments are difficult. However, Hernando-Espinilla et al. [34], using a cohort of 41 healthy older adults (60–90 years), presented some standards. The reference value for MDA was 0.23 ± 0.01 nmol/mg, while the GSH/GSSG ratio was set at 1.25 ± 0.33% [34]. Minimal differences between these statements and our study outcomes could be due to the dissimilarities in participants’ age and sex.

We encountered the same positive trend for lipid peroxidation (i.e., MDA) as one previous study [35]. In this study [35], an RT program of 24 weeks (3 days per week, 12 exercises, low intensity: one set of 13 repetitions at 50% 1RM; high intensity: one set of eight repetitions at 80% 1RM) significantly reduced MDA in elderly men and women (14% reduction in low intensity and 18% in high intensity). However, in our study, significant differences were not achieved. This disagreement can be attributed to the higher training volume used in our study (three to four sets vs. one set), which may have unbalanced the REDOX state [36]. Concerning the effects of RT on GSH/GSSG, our results are in contrast to those of Peters et al. [37], who reported that GSH/GSSH was increased (Δ% = +61%) in hypertensive adults after six weeks of isometric training. Our findings also disagree with Çakir-Atabek et al. [38], who found significant improvements in antioxidant enzymes after six weeks of RT (70–85% 1RM). In this sense, although, in our study, GSH/GSSG remained within healthy older adults’ cut-off points, significant differences between pre- and post-intervention were not reported. Concerning the effects of RT on IL-6, our results agree with previous research [39] that encountered a chronic reduction of IL-6 levels with aerobic and RT. Hence, regardless of the type of physical exercise performed, it seems that the intensity, duration, devices employed, characteristics, and physical condition of the participants, as well as the inflammation grade are key factors in the management of oxidative cytokines.

Finally, the magnesium-rich supplementation did not significantly improve vitamin D levels when baseline vitamin D values were low [40]. Magnesium supplementation only increased vitamin D values when vitamin D baseline levels were at 30 ng/mL, but not at 20 ng/mL [40]. In contrast, we found that RT + SW obtained significant improvements in vitamin D concentration when baseline levels were at 20 ng/mL (RT + SWpre = 25.56 ng/mL; RT + SWpost = 29.06 ng/mL). These results could be beneficial for improving antioxidant function in the study population, as vitamin D is involved in mitochondrial function and calcium homeostasis and, additionally, is an antioxidant that could prevent iron-dependent lipid peroxidation in the cell membrane [41].

### 4.3. Systolic and Diastolic Blood Pressure Responses

Mineral ingestion (i.e., Mg^2+^, Ca^2+^, and Na^+^) might influence the physiological regulation of vascular tone, which might reduce the risk of atherosclerosis and high blood pressure [42]. Accordingly, we encountered significant reductions in SBP and DBP after 32 weeks of RT with EB (SBP, RT + SW: *p* < 0.001; ES = 0.71; RT + PLA: *p* < 0.001; ES = 0.79; DBP, RT + SW: *p* = 0.002; ES = 0.51; RT + PLA: *p* = 0.104; ES = 0.27). Similarly, CON + SW had their DBP reduced (*p* = 0.028; ES = 0.50). The interindividual analysis conducted in the SBP confirms the similar ES obtained in both RT + SW and RT + PLA. In humans, multimineral supplementation has shown greater improvements in hypertensive (weighted mean difference [WMD]: −7.98 mmHg, 95%CI [−14.95 to −1.02]) compared to healthy subjects (WMD: −1.25 mmHg, 95%CI [−2.48 to −0.02]) [43]. However, when RT interventions are examined, blood pressure is also significantly decreased in normotensive (WMD: −3.9 mmHg, 95%CI [−6.40 to −1.20]) and prehypertensive subjects (WMD: −3.9 mmHg, 95%CI [−5.6 to −2.2]) [24]. In our study, according to the reference’s values, participants can be defined as prehypertensive [44]. Therefore, this 32-week intervention elicited higher changes than those expected in normotensive patients. Overall, this is the first study measuring blood pressure after SW ingestion in the long term. Despite the promising results, even when SW is consumed in the CON (see Table 3), further studies are needed to confirm the adjuvant effect of adding a mineral-enriched supplement to RT.

### 4.4. Limitations, Strengths, and Future Research

To our knowledge, this is the first study confirming the effects of SW supplementation and RT with EB during 32 weeks in 93 postmenopausal women. Considering the acute benefits of SW (e.g., lactate buffer) [7], it would be interesting to combine the analyzed chronic adaptations with potential acute benefits (e.g., number of repetitions, rate of perceived exertion). Nevertheless, this research is not free of limitations. First, an exhaustive dietary and lifestyle registry was not carried out. In this regard, it is known that diet, sun exposure [45], alcohol, and antioxidant ingestion [46] may alter hepatic, inflammatory, and oxidative stress values. Thus, it is mandatory to control these variables in future studies. Besides that, future research could further investigate the long-term potential effects of this supplement on health and performance in subjects suffering from certain conditions.

## 5. Conclusions

We can state that the intake of SW did not elicit hepatic damage in any of the groups. Indeed, GPT values improved in both RT groups (with and without SW supplementation). While oxidative stress values may not vary with long-term interventions, a mineral-enriched supplement added to a 32-week RT program improves vitamin D status and inflammatory profile (reducing the IL-6 levels). Moreover, SBP improved in both RT groups and DBP improved in both groups with SW supplementation (performing and not performing RT). Therefore, SW supplementation added to RT or alone could be effective at improving blood pressure in older women without affecting hepatic and oxidative stress status. Additionally, SW supplied in combination with RT positively influences vitamin D levels and inflammatory status.

## Figures and Tables

**Figure 1 healthcare-12-00204-f001:**
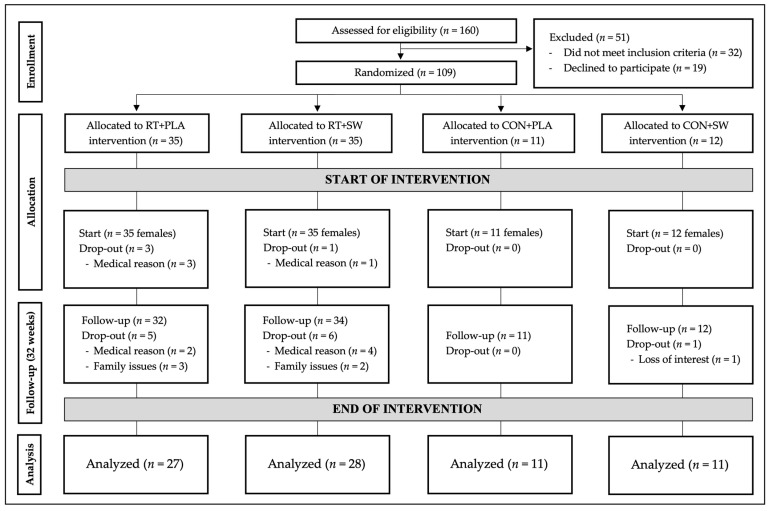
Flow of participants throughout the study. RT: resistance training; PLA: placebo; SW: seawater supplement; CON: control group. Participants are presented according to previous studies of this same project [25].

**Figure 2 healthcare-12-00204-f002:**
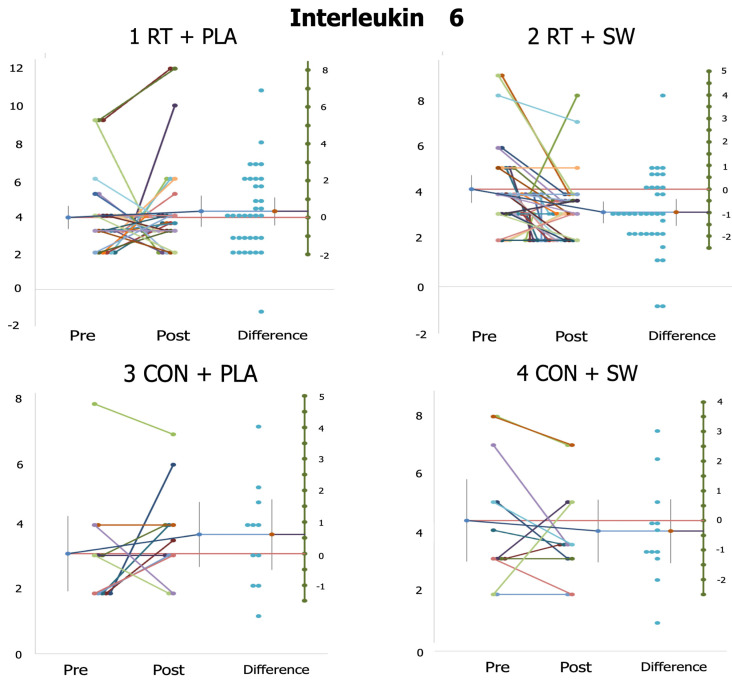
Individual responses of participants from each of the study groups in terms of interleukin 6. Colored shorter lines indicate the change from pre- to post-intervention of each participant. The longer line (which presents error bars on its extremes) indicates the mean change of the group. The red lines indicate the mean pre-intervention value of the group (this value is standardized to zero on the green column of the right-hand side of each plot). The blue lines indicate the mean post-intervention value of the group, which is connected to a red point that indicates the mean change from pre- and post-intervention of each group. Blue points on the right-hand side represent the difference obtained by each participant. RT: resistance training; PLA: placebo; SW: seawater supplement; CON: control group.

**Figure 3 healthcare-12-00204-f003:**
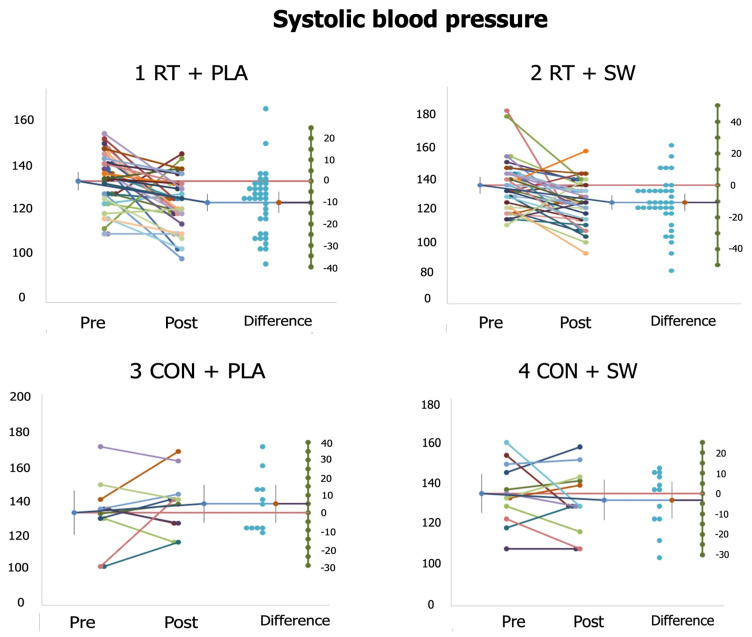
Individual responses of participants from each of the study groups in terms of systolic blood pressure. Colored shorter lines indicate the change from pre- to post-intervention of each participant. The longer line (which presents error bars on its extremes) indicates the mean change of the group. The red lines indicate the mean pre-intervention value of the group (this value is standardized to zero on the green column of the right-hand side of each plot). The blue lines indicate the mean post-intervention value of the group, which is connected to a red point that indicates the mean change from pre- and post-intervention of each group. Blue points on the right-hand side represent the difference obtained by each participant. RT: resistance training; PLA: placebo; SW: seawater supplement; CON: control group.

**Table 1 healthcare-12-00204-t001:** Baseline descriptive characteristics of the participants.

	1 RT + PLA (*n* = 35)	2 RT + SW (*n* = 35)	3 CON + PLA (*n* = 11)	4 CON + SW (*n* = 12)
Age (years)	69.17 ± 5.71	70.80 ± 5.86	67.90 ± 8.60	72.00 ± 7.07
Weight (kg)	66.25 ± 9.41	68.25 ± 11.41	64.36 ± 6.80	71.67 ± 10.46
Height (cm)	152.63 ± 4.54	153.46 ± 5.97	150.95 ± 5.19	153.24 ± 5.35
BMI (kg/m^2^)	28.41 ± 3.99	28.83 ± 4.73	28.50 ± 3.57	29.88 ± 4.69
Fat mass (%)	43.15 ± 1.39	43.94 ± 4.56	41.66 ± 3.26	45.91 ± 5.43
UGT (s)	6.74 ± 1.00	6.79 ± 0.86	6.39 ± 1.61	6.38 ± 0.87

BMI: body mass index; UGT: up-and-go-test. RT: resistance training; PLA: placebo; SW: microfiltered sea water; CON: control group. Participants are presented according to previous studies of this same project [25].

**Table 2 healthcare-12-00204-t002:** Renal and hepatic biomarkers.

	1 RT + PLA (*n* = 35)		2 RT + SW (*n* = 35)		3 CON + PLA (*n* = 11)		4 CON + SW (*n* = 12)	
	Pre	Post	Pre	Post	Pre	Post	Pre	Post
GOT	22.23 ± 6.24	21.42 ± 4.77	21.20 ± 5.76	19.81 ± 3.76	19.80 ± 3.58	20.39 ± 3.69	24.09 ± 4.30	22.79 ± 6.27
GPT	19.68 ± 8.54	17.02 ± 6.23 * d = 0.36	19.44 ± 7.57	16.20 ± 4.68 * d = 0.51	17.70 ± 7.79	18.40 ± 6.02	18.73 ± 6.31	17.90 ± 4.18
GGT	21.26 ± 13.98	22.36 ± 13.11	20.09 ± 11.19	18.84 ± 7.50	24.00 ± 15.89	23.79 ± 12.90	20.36 ± 8.90	19.98 ± 5.40
ALP	73.97 ± 17.67	73.36 ± 13.63	66.32 ± 16.43	65.46 ± 11.97	71.30 ± 15.23	71.61 ± 12.81	68.64 ± 17.69	72.32 ± 12.95

RT: resistance training; SW: seawater supplement; CON: control group; GOT: glutamic-oxaloacetic transaminase; GPT: glutamic-pyruvic transaminase; GGT: gamma-glutamyl transpeptidase; ALP: alkaline phosphatase enzyme. * Significant differences (*p* < 0.05) between the preintervention and postintervention measurements. d: Cohen’s d effect size of the significant differences.

**Table 3 healthcare-12-00204-t003:** Oxidative stress biomarkers, inflammatory biomarkers, and vitamin D.

	1 RT + PLA (*n* = 35)		2 RT + SW (*n* = 35)		3 CON + PLA (*n* = 11)		4 CON + SW (*n* = 12)	
	Pre	Post	Pre	Post	Pre	Post	Pre	Post
MDA (nmol/mg)	0.19 ± 0.08 ^4^	0.19 ± 0.04 ^4^	0.20 ± 0.06	0.17 ± 0.06 ^4^	0.18 ± 0.07	0.18 ± 0.07 ^4^	0.26 ± 0.05	0.26 ± 0.05
GSH/GSSG (%)	1.18 ± 0.44	1.21 ± 0.46	1.08 ± 0.37	1.21 ± 0.46	1.25 ± 0.75	1.25 ± 0.75	0.97 ± 0.24	0.98 ± 0.25
Vitamin D (ng/mL)	26.31 ± 11.66	28.63 ± 13.59	25.56 ± 13.56	29.06 ± 13.99 * d = 0.25	23.10 ± 7.92	22.40 ± 7.37	17.73 ± 11.15	20.27 ± 10.19
IL-6 (pg/mL)	3.91 ± 1.84	4.25 ± 2.45	4.03 ± 1.64	3.03 ± 1.22 *^,1^ d = 0.69	2.70 ± 0.82	3.47 ± 1.17	4.09 ± 1.97	3.78 ± 1.45

RT: resistance training; SW: seawater supplement; CON: control group; MDA: malondialdehyde; GSH/GSSG: oxidized/reduced glutathione ratio; IL-6: interleukin 6. * Significant differences (*p* < 0.05) between the pre-intervention and pos-tintervention measurements. ^1,4^ Significant differences (*p* < 0.05) with Groups 1 or 4, respectively. d: Cohen’s d effect size of the significant differences.

**Table 4 healthcare-12-00204-t004:** Systolic and diastolic blood pressure.

	1 RT + PLA (*n* = 35)		2 RT + SW (*n* = 35)		3 CON + PLA (*n* = 11)		4 CON + SW (*n* = 12)	
	Pre	Post	Pre	Post	Pre	Post	Pre	Post
SBP (mmHg)	134.01 ± 12.75	124.11 ± 12.31 *^,3^ d = 0.79	136.10 ± 16.07	125.43 ± 13.69 *^,3^ d = 0.71	132.55 ± 19.99	139.40 ± 15.29	133.77 ± 15.70	131.41 ± 15.57
DBP (mmHg)	75.60 ± 7.02	73.46 ± 8.65	76.23 ± 7.91	71.91 ± 9.04 *^,3^ d = 0.51	78.95 ± 10.87	81.40 ± 9.10	82.09 ± 11.64	76.82 ± 9.10 * d = 0.50

RT: resistance training; SW: seawater supplement; CON: control group; SBP: systolic blood pressure; DBP: diastolic blood pressure. * Significant differences (*p* < 0.05) between the preintervention and postintervention measurements. ^3^ Significant differences (*p* < 0.05) with Group 3. d: Cohen’s d effect size of the significant differences.

## Data Availability

Data are available upon request to the corresponding author.

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
