# Peer review of "Long-Term Effects of Microfiltered Seawater and Resistance Training with Elastic Bands on Hepatic Parameters, Inflammation, Oxidative Stress, and Blood Pressure of Older Women: A 32-Week, Double-Blinded, Randomized, Placebo-Controlled Trial"

_healthcare, 2024, doi:10.3390/healthcare12020204_

Round 1

Reviewer 1 Report

Comments and Suggestions for Authors

This clinical trial explores the effects of resistance training (RT) and sea water (SW) on several biochemical, inflammatory and vascular parameters in a post-menopausal women population.

In general I see a wonderful long-term work with very nice results, however, in order to improve the scientific soundness I believe several issues must be addressed, above all, in terms of design, results presentation and discussion

1.-To motivate participants to maintain their usual life habits and control it by oral communication is not enough. There are several aspects affecting the measured parameters: alcohol consumption, sun exposure due to a trip, antioxidant consumption through wine, chocolate. I believe, this should have been controled in a closer and more systematic way in order to take it into consideration during analysis. If this information is available I highly recommend to present it with detail, and if not, to discuss it.

2.- Dietary evaluation together with biochemical parameters would have been nice, at least also to see that dietary parameters remained the same grosso modo during the trial. Please discuss about this

3.- Since 8 weeks are a very long period, comparing pre and post withouth check points in the middle can be tricky. I would have appreciated at least one checkpoint in the middle (month 4) in order to see the behaviour of parameters. If this is not possible, please discuss about it.

4.- Anova analysis is necessary to see if the parameters that are significant remain significant after adjustment of variables.

5.- Line 311 says: "enhanced IL-6........ values". I would change it to "improved IL-6......... values"

6.- SW composition should be described in the Methods section

7.- Papers using SW differentiate the use of SW and deep sea water, please explain if this is Deep sea water (this is not mentioned in the manufacturers webpage) and if not, what is the differences in terms of content and supplementation effects.

8.- Since placebo was natural water, I highly suspect that participants did recognize in which group they were included, because SW had a salty taste. Did you confirm this or ask about this? This could bias some effects above all blood pressure responses.

9.- Figures need to be improved in image definition

10.- Figure 2 is very poorly described, this needs to be expaned and above all, to highlight the importance of presenting individual responses. Also to mention this in the discussion.

Author Response

Review healthcare-2757623

Could microfiltered-seawater supplementation and variable resistance training improve hepatic, inflammatory, and oxidative stress profiles along with blood pressure in older women? A 32-week, double-blinded, randomized, placebo-controlled trial

Reviewer 1

This clinical trial explores the effects of resistance training (RT) and sea water (SW) on several biochemical, inflammatory and vascular parameters in a post-menopausal women population. In general, I see a wonderful long-term work with very nice results, however, in order to improve the scientific soundness, I believe several issues must be addressed, above all, in terms of design, results presentation and discussion.

Response: We thank the reviewer for the valuable assessment of our article. We have answered to all the issues raised by the reviewer and modified the manuscript accordingly. Please, find all the amendments performed to the revised version of the manuscript highlighted with the track changes function.

1.-To motivate participants to maintain their usual life habits and control it by oral communication is not enough. There are several aspects affecting the measured parameters: alcohol consumption, sun exposure due to a trip, antioxidant consumption through wine, chocolate. I believe, this should have been controlled in a closer and more systematic way in order to take it into consideration during analysis. If this information is available, I highly recommend to present it with detail, and if not, to discuss it.

Response: Thank you for your comment and for pointing out important aspects to consider in our study. It is true that controlling and recording all factors that could influence the measured parameters is crucial. We want to emphasize that the limitation mentioned by our reviewer is relevant, so much so that in other studies posteriorly conducted by our research group, we controlled for it through a self-reported diet record over three consecutive days at the beginning and the end of the training program (Gargallo et al., 2023). We acknowledge the significance of this point and, therefore, we have mentioned it within the study's limitations for a more comprehensive evaluation of the outcomes. Thank you.

Sentence added to the Limitations Section: “Nevertheless, this research is not free of limitations. First, an exhaustive dietary and lifestyle registry was not carried out. In this regard, it is known that diet, sun exposure [44], alcohol, and antioxidant ingestion [45] may alter hepatic, inflammatory and oxidative stress values. Thus, it is mandatory to control these variables in future studies.”

2.- Dietary evaluation together with biochemical parameters would have been nice, at least also to see that dietary parameters remained the same grosso modo during the trial. Please discuss about this

Response: Thank you for the comment once again. As mentioned in the previous section, a specific dietary registry was not conducted. Participants were instructed not to alter their dietary or physical activity habits during the study. Simultaneously, patients were informed about the protocol to be followed during assessments, preventing biochemical results from being affected by acute factors such as medication intake, alcohol, supplements, or intense physical exercise. As aforementioned, we have added in Limitations Section a paragraph highlighting what our reviewer points out in Questions 1 and 2.

3.- Since 8 weeks are a very long period, comparing pre and post without check points in the middle can be tricky. I would have appreciated at least one checkpoint in the middle (month 4) in order to see the behaviour of parameters. If this is not possible, please discuss about it.

Response: Thank you again for your comment. In this case, no additional assessments were conducted for various reasons. The primary constraint was logistical; given the study's sample size, the number of variables analyzed, available space, and human resources, it was quite challenging to conduct more assessments than those performed. Additionally, based on previous studies and physiological mechanisms, we considered an 8-month duration as optimal to observe potential adaptations resulting from the exercise performed. Therefore, considering the cost-benefit ratio, this time frame was decided upon as the most appropriate.

4.- Anova analysis is necessary to see if the parameters that are significant remain significant after adjustment of variables.

Response: We thank our reviewer for this suggestion. According to previous literature (Zhang et al., 2014), we conducted a two-way mixed ANOVA considering that there were no significant differences in the baseline in most of the dependent variables. However, bearing in mind that we encountered significant differences between RT+PLA and CON+SW in the baseline levels of malondialdehyde we have conducted an ANCOVA to adjust for the differences in the baseline levels. In this regard and as shown below, we have found the same results (i.e., significant differences between CON+SW and the rest of the study groups).

Mean Difference

Std. Error

Sig.b

95% Confidence Interval

Lower Bound

Upper Bound

RT+ PLA

RT+SW

.014

.012

.261

-.010

.038

CON+PLA

.004

.018

.815

-.032

.041

CON+SW

-.053*

.019

.005

-.090

-.016

RT+SW

CON+PLA

-.010

.019

.609

-.046

.027

CON+SW

-.067*

.018

<.001

-.104

-.030

CON+PLA

CON+SW

-.058*

.023

.016

-.104

-.011

5.- Line 311 says: "enhanced IL-6........ values". I would change it to "improved IL-6......... values"

Response: Amended. Thank you!

6.- SW composition should be described in the Methods section

Response: We appreciate this comment aimed at improving the clarity of our manuscript. We have reworded the sentence in methods section:

“The mineral-enriched supplement was composed of 11.87 g L−1 sodium; 20.36 g L−1 chloride; 1.36 g L−1 magnesium; 433 mg L−1 calcium; 441 mg L−1 potassium; 148 mg L−1 bicarbonate; 11.8 μg L−1 zinc; 116.9 μg L−1 manganese; and 6.6 μg L−1 cupper. Moreover, other chemical components of seawater were present in the supplement in trace magnitudes: proteins, lipids, and water- and fat-soluble vitamins (D-biotin, thiamine, riboflavin, nicotinamide, cyanocobalamin, pyridoxine, retinal, D3, α-tocopherol, and K1).”

7.- Papers using SW differentiate the use of SW and deep sea water, please explain if this is Deep sea water (this is not mentioned in the manufacturers webpage) and if not, what is the differences in terms of content and supplementation effects.

Response: Thank you for giving us the possibility to further discuss this relevant issue. Unfortunately, the current literature is not accurate enough in the term used. The terms sea water (SW), deep-sea water (DSW), deep-mineral water (DMW), and deep-ocean mineral water (DOMW) are usually used indistinctly (Aragón-Vela et al., 2022). However, it is difficult to establish a clear definition for each one since the salinity and water characteristics of the oceans differ between them (Stasiule et al., 2014; Keen et al., 2010; González-Acevedo et al., 2022). Therefore, the composition of the aforementioned supplement can vary at the same depth in different oceans.

Regarding the laboratory that supplied us the supplement, we can state that this water has been extracted from specific areas of the Atlantic Ocean, in the Bay of Biscay, at more than 10 miles from the coast and 20 meters depth, at the center of proliferations or plankton blooms. Once the water is extracted and, to guarantee the cold chain, it is transported to the laboratories at low temperatures. Once at the laboratory, after an initial physical-chemical and microbiological analysis, it follows a double cold microfiltration process in a white room, respecting the regulations established by the European Pharmacopoeia. Immediately after, the finished product is reanalyzed and packaged under aseptic conditions.

8.- Since placebo was natural water, I highly suspect that participants did recognize in which group they were included, because SW had a salty taste. Did you confirm this or ask about this? This could bias some effects above all blood pressure responses.

Response: Thank you again for the possibility to further discuss this relevant issue. In this case, this situation was avoided by arranging the elderly individuals across different centers and time sessions in a way that the various groups with different concentrated serum and water never coincided in the same center or group. Therefore, they did not know the actual differences between the supplement or the placebo. Furthermore, the blood pressure was not significantly different in CON+PLA and CON+SW. Therefore, it seems that blood pressure was not significantly modified by the intake of the supplement or the placebo. We remain at your disposal for further discussion in this regard. Thank you!

9.- Figures need to be improved in image definition

Response: Thank you for this suggestion. We have now attached the images in separated files (.tiff and .png) to improve resolution.

10.- Figure 2 is very poorly described, this needs to be expaned and above all, to highlight the importance of presenting individual responses. Also to mention this in the discussion.

Response: Thank you for pointing out this relevant issue. We have taken into consideration your advice and thoroughly described the figures. Additionally, we have interpreted the results of the figure in Results and Discussion Sections, as you can see in the revised version of the manuscript.

Additional references cited

Aragón-Vela J, González-Acevedo O, Plaza-Diaz J, Casuso RA, Huertas JR. Physiological Benefits and Performance of Sea Water Ingestion for Athletes in Endurance Events: A Systematic Review. Nutrients. 2022;14(21):4609. doi:10.3390/nu14214609

Gargallo, P., Tamayo, E., Jiménez-Martínez, P., Juesas, A., Casaña, J., Benitez-Martinez, J. C., Gene-Morales, J., Fernandez-Garrido, J., Saez, G. T., & Colado, J. C. (2023). Multicomponent and power training with elastic bands improve metabolic and inflammatory parameters, body composition and anthropometry, and physical function in older women with metabolic syndrome: A 20-week randomized, controlled trial. Experimental Gerontology, 112340. https://doi.org/10.1016/j.exger.2023.112340

Gonzalez-Acevedo O., Aragon-Vela J., De la Cruz Marquez J.C., Marin M.M., Casuso R.A., Huertas J.R. Seawater Hydration Modulates IL-6 and Apelin Production during Triathlon Events: A Crossover Randomized Study. Int. J. Environ. Res. Public Health. 2022;19:9581. doi: 10.3390/ijerph19159581

Keen D.A., Constantopoulos E., Konhilas J.P. The impact of post-exercise hydration with deep-ocean mineral water on rehydration and exercise performance. J. Int. Soc. Sports Nutr. 2016;13:17. doi: 10.1186/s12970-016-0129-8

Stasiule L., Capkauskiene S., Vizbaraite D., Stasiulis A. Deep mineral water accelerates recovery after dehydrating aerobic exercise: A randomized, double-blind, placebo-controlled crossover study. J. Int. Soc. Sports Nutr. 2014;11:34. doi: 10.1186/1550-2783-11-34

Zhang, S., Paul, J., Nantha-Aree, M., Buckley, N., Shahzad, U., Cheng, J., DeBeer, J., Winemaker, M., Wismer, D., Punthakee, D., Avram, V., & Thabane, L. (2014). Empirical comparison of four baseline covariate adjustment methods in analysis of continuous outcomes in randomized controlled trials. Clinical Epidemiology, 6, 227-235. https://doi.org/10.2147/CLEP.S56554

Reviewer 2 Report

Comments and Suggestions for Authors

The Figures are very low resolution.

2023 only 1 reference need to cite atleast 10.

Example

Lee SR, Directo D. Fish Oil Supplementation with Resistance Exercise Training Enhances Physical Function and Cardiometabolic Health in Postmenopausal Women. Nutrients. 2023 Oct 25;15(21):4516. doi: 10.3390/nu15214516. PMID: 37960168; PMCID: PMC10650161.

Juesas, A.; Gargallo, P.; Gene-Morales, J.; Babiloni-López, C.; Saez-Berlanga, A.; Jiménez-Martínez, P.; Casaña, J.; Benitez-Martinez, J.C.; Ramirez-Campillo, R.; Chulvi-Medrano, I.; et al. Effects of Microfiltered Seawater Intake and Variable Resistance Training on Strength, Bone Health, Body Composition, and Quality of Life in Older Women: A 32-Week Randomized, Double-Blinded, Placebo-Controlled Trial. Int. J. Environ. Res. Public Health 202320, 4700. https://doi.org/10.3390/ijerph20064700

Advised not  to submit same article in different journals same time.

Author Response

Reviewer 2

The Figures are very low resolution.

Response: Thank you for this suggestion. We have now attached the images in different files (.tiff and .png) to improve resolution.

2023 only 1 reference need to cite at least 10.

Example: Lee SR, Directo D. Fish Oil Supplementation with Resistance Exercise Training Enhances Physical Function and Cardiometabolic Health in Postmenopausal Women. Nutrients. 2023 Oct 25;15(21):4516. doi: 10.3390/nu15214516. PMID: 37960168; PMCID: PMC10650161.

 Response: Thank you for your comment aimed at improving the quality of our manuscript. We have removed some references and added more novel literature in order to not exceed the number of citations. Due to our reference’s changes, we have cited 10 articles published between 2022 and 2023. And almost the half of our citations are based on articles published the last five years (from 2019 to 2023).

Juesas, A.; Gargallo, P.; Gene-Morales, J.; Babiloni-López, C.; Saez-Berlanga, A.; Jiménez-Martínez, P.; Casaña, J.; Benitez-Martinez, J.C.; Ramirez-Campillo, R.; Chulvi-Medrano, I.; et al. Effects of Microfiltered Seawater Intake and Variable Resistance Training on Strength, Bone Health, Body Composition, and Quality of Life in Older Women: A 32-Week Randomized, Double-Blinded, Placebo-Controlled Trial. Int. J. Environ. Res. Public Health 202320, 4700. https://doi.org/10.3390/ijerph20064700

Advised not to submit same article in different journals same time.

Response: Thank you for this suggestion. We kindly ask the reviewer to consider the different topics reported in each manuscript. The aforementioned manuscript explains how this mineral-enriched supplement can influence physical parameters (isokinetic strength, bone mineral density, and body composition) and quality of life. Nevertheless, this second part evaluates adaptations in physiological parameters (hepatic, inflammatory, and oxidative stress profiles, and blood pressure). This was done to resolve the existing controversy regarding the high amount of sodium that contains microfiltered seawater. Since sodium excess has been commonly associated with alterations in the homeostatic physiology of the body (including hepatic parameters, oxidative stress, inflammation, and blood pressure among others), especially in older adults.

In summary, these two studies derived from the same project diverge in terms of the fields of application. On the one hand, the first study would be more relevant for professionals focused on the prevention of older adult falls, improvement of quality of life, and/or rehabilitation (e.g., Sports Scientists, Physiotherapists, Psychologists, and Physicians). On the other hand, the second study could be found interesting by professionals more focused on physiological health (e.g., Nutritionists, Hepatologists, Endocrinologists, Medical Doctors, apart from the aforementioned).

Reviewer 3 Report

Comments and Suggestions for Authors

Dear Editor,

The manuscript by Babiloni-López et al., Could microfiltered-seawater supplementation and variable resistance training improve hepatic, inflammatory, and oxidative stress profiles along with blood pressure in older women? A 32-week, double-blinded, randomized, placebo-controlled trial describes a study focused on analyzing the impact of elastic bands (EBs) resistance training (RT) program (32 weeks) combined with microfiltered seawater (SW) intake on hepatic biomarkers, oxidative stress, and blood pressure in post-menopausal women. The manuscript is stepwise structured and the experiments are carefully designed to answer the objectives and test their set hypotheses.

I have carefully read through the manuscript and have these comments to make to improve on understanding and add clarity.

1.     The title looks quite wordy and could be rephrased.

2.     In the Abstract, the authors fail to provide any background but dive directly into the aim of the study. This background is needed to provide proper context. Also, the use of non-standard abbreviations like PLA in the Abstract and TBA in the Methodology should be corrected.

3.     Also, the authors fail to give any implications, impact, or potential exploitation of their results in the Abstract.

4.     In the Introduction, the authors fail to lay a proper background on the study, especially on the choice of the study population,

5.     On the experimental design, I am not certain about the authors’ decision to conceal the names of institutions, IRBs, and others for peer-review reasons.

6.     Lines 180-181 read, “Serum separation involved drawing 10 whole blood samples into dry tubes with a gel separator and coagulation activator”. This should be looked at and corrected. This goes for the sentence in lines 58-60.

7.     In the Results (3.1), the authors should provide information on the number of participants that ended up in each group in the final analyses.

8.     The tables in the Results section should be created to fit on a single page. Also, in presenting the results, I think it is better to talk about the results and reference to a table instead of the approach used by the authors.

9.     In the last part of the Results where the authors talk about weak adverse events, they should include the number of participants who manifested any of these.

10.  In the Discussion, the authors should begin by clearly stating the problem they set out to solve before proceeding.

11.  Lastly, the paper has grammatical errors in so many places which sometimes obscure the meaning and makes understanding difficult. It will be important for the authors to do proper proofreading.

Comments on the Quality of English Language

Can be improved

Author Response

Reviewer 3

Dear Editor,

 The manuscript by Babiloni-López et al., Could microfiltered-seawater supplementation and variable resistance training improve hepatic, inflammatory, and oxidative stress profiles along with blood pressure in older women? A 32-week, double-blinded, randomized, placebo-controlled trial describes a study focused on analyzing the impact of elastic bands (EBs) resistance training (RT) program (32 weeks) combined with microfiltered seawater (SW) intake on hepatic biomarkers, oxidative stress, and blood pressure in post-menopausal women. The manuscript is stepwise structured and the experiments are carefully designed to answer the objectives and test their set hypotheses.

 I have carefully read through the manuscript and have these comments to make to improve on understanding and add clarity.

Response: We thank the reviewer for the time spent reading and reviewing our article. We have considered all the suggestions for the improvement of our manuscript. Please, find made in the revised version are highlighted with the track changes function of Microsoft Word software. 

  1. The title looks quite wordy and could be rephrased.

Response: Thank you for this suggestion. We have reworded the title to make it shorter and concise.

  1. In the Abstract, the authors fail to provide any background but dive directly into the aim of the study. This background is needed to provide proper context. Also, the use of non-standard abbreviations like PLA in the Abstract and TBA in the Methodology should be corrected.

Response: A short introductory background is provided now. Besides that, the abbreviations have been removed. Thank you!

“The bulk of research on microfiltered seawater (SW) is mainly based on its short-term effects. However, the long-term physiological adaptations to combining SW and resistance training (RT) are unknown.”

  1. Also, the authors fail to give any implications, impact, or potential exploitation of their results in the Abstract.

Response: A short field implication is provided now. Thank you!

“Therefore, RT+SW or SW alone are safe strategies in the long term with no influences on hepatic and oxidative stress biomarkers. Additionally, SW in combination with RT positively influences vitamin D levels, inflammation, and blood pressure in older women.”

  1. In the Introduction, the authors fail to lay a proper background on the study, especially on the choice of the study population,

Response: Thank you once again for the comment. We agree with you! Older adults were our target population due to the fact that elevated blood pressure and restriction or reduction in high-sodium diets are commonly seen in this kind of population. We have now included some phrases to clarify why we chose this population.

“However, due to the high levels of sodium that SW contains, its beneficial impact on older adults should be cautiously interpreted [8]. Senescent changes lead to an increase in hypertension due to vascular resistance, which can impair the pump activity of the sodium membrane [9]. Therefore, high-sodium diets may increase blood pressure and risk for liver and kidney diseases [10]. Considering that no long-term studies about the benefits or risks of SW on hepatic biomarkers have been conducted in older adults, novel research approaching this topic may be valuable.”

  1. On the experimental design, I am not certain about the authors’ decision to conceal the names of institutions, IRBs, and others for peer-review reasons.

Response: As a research group, we tend to conceal as much particular names and places as possible in order to not bias, mainly, the editor decision. Besides that, we believe that the concealed names do not influence the meaning and clarity of the manuscript. Thank you for your consideration!

  1. Lines 180-181 read, “Serum separation involved drawing 10 whole blood samples into dry tubes with a gel separator and coagulation activator”. This should be looked at and corrected. This goes for the sentence in lines 58-60.

Response: Indeed, it was a typo. Checked and amended. Thank you!

“Serum separation involved drawing 10 mL whole blood samples into dry tubes with a gel separator and coagulation activator”.

  1. In the Results (3.1), the authors should provide information on the number of participants that ended up in each group in the final analyses.

Response: We appreciate this comment aimed at improving the clarity of our manuscript. Initially and as our reviewer can see, we presented this information in Figure 1, where the flow chart of participants is shown. In the figure is clearly explained the drop-out of participants and how much participants ended up the intervention in each group (RT+PLA: 27; RT+SW: 28; CON+PLA: 11; CON+SW: 11). However, as our reviewer suggests we have added this information in 3.1 Participants. Thank you!

  1. The tables in the Results section should be created to fit on a single page. Also, in presenting the results, I think it is better to talk about the results and reference to a table instead of the approach used by the authors.

Response: We appreciate this suggestion aimed at increasing the reach and soundness of our manuscript. To better fit our tables and figures in the revised version of the manuscript, we have modified Figure 2 into Figures 2 and 3. Additionally, we have rearranged the tables where necessary to fit them in one page each. Regarding the results, as our reviewer can see in the revised version of the manuscript, we have explained in a subtle way the post-hoc comparisons referencing the table. With these amendments we believe that the results can be now easily followed and that they are more understandable and comprehensible than the previous version. Thank you for your advice!

  1. In the last part of the Results where the authors talk about weak adverse events, they should include the number of participants who manifested any of these.

Response: Thank you once again for the comment. As indicated in the manuscript, the only adverse effect recorded was muscle soreness, which was reported by a total of 23 subjects, mainly in the initial weeks.

  1. In the Discussion, the authors should begin by clearly stating the problem they set out to solve before proceeding.

Response: Thank you for your suggestion. We have reworded the first lines of the discussion to report our mainly concern when the article was conceived.

“To the best of our knowledge, this is the first study to demonstrate that SW intake added to RT or alone has no adverse effects on hepatic and inflammation parameters in the long-term”.

  1. Lastly, the paper has grammatical errors in so many places which sometimes obscure the meaning and makes understanding difficult. It will be important for the authors to do proper proofreading.

Response: We appreciate this comment aimed at improving the clarity of our manuscript. We have double-checked all the manuscript to correct grammatical errors in the revised version of the manuscript. As our reviewer can see there are a considerably large number of modifications that improve the grammatical correctness of our manuscript. Thank you!

Round 2

Reviewer 1 Report

Comments and Suggestions for Authors

I am pleased with the response and changes. Thanks

Author Response

Reviewer 1

I am pleased with the response and changes. Thanks

Response: We appreciate your time and positive comments. We believe the article is now much improved. At your disposal!

Reviewer 2 Report

Comments and Suggestions for Authors

Tables data and Flow chart is same as published paper, even the outcome different parameter it is not recommended

Juesas, A.; Gargallo, P.; Gene-Morales, J.; Babiloni-López, C.; Saez-Berlanga, A.; Jiménez-Martínez, P.; Casaña, J.; Benitez-Martinez, J.C.; Ramirez-Campillo, R.; Chulvi-Medrano, I.; et al. Effects of Microfiltered Seawater Intake and Variable Resistance Training on Strength, Bone Health, Body Composition, and Quality of Life in Older Women: A 32-Week Randomized, Double-Blinded, Placebo-Controlled Trial. Int. J. Environ. Res. Public Health 202320, 4700. https://doi.org/10.3390/ijerph20064700

Author Response

Reviewer 2

Tables data and Flow chart is same as published paper, even the outcome different parameter it is not recommended

Juesas, A.; Gargallo, P.; Gene-Morales, J.; Babiloni-López, C.; Saez-Berlanga, A.; Jiménez-Martínez, P.; Casaña, J.; Benitez-Martinez, J.C.; Ramirez-Campillo, R.; Chulvi-Medrano, I.; et al. Effects of Microfiltered Seawater Intake and Variable Resistance Training on Strength, Bone Health, Body Composition, and Quality of Life in Older Women: A 32-Week Randomized, Double-Blinded, Placebo-Controlled Trial. Int. J. Environ. Res. Public Health 2023, 20, 4700. https://doi.org/10.3390/ijerph20064700

Response: Thank you again for reviewing our manuscript and letting us explain this concern.

We believe it was necessary to publish two articles from the same main project considering that both present different theoretical frameworks and provide the scientific body of knowledge with relevant information from two different points of view (one more focused on neuromuscular aspects and the other on metabolic and biochemicals). Furthermore, the main dataset produced in the project is too large to be reported in a single publication, and this is a valid reason to divide a project into two publications (Altay & Koçak, 2021). In the mentioned article (Altay & Koçak, 2021), the authors identify an ethical concern “the publication of a paper that overlaps substantially with one already published, without clear, visible reference to the previous publication.” We believe that both manuscripts do not conflict with the item mentioned as they only present the same samples but different dependent variables and outcomes, as previously performed by published expert literature (Franzke et al., 2018, 2019). Furthermore, in the first manuscript published (Juesas et al., 2023), we cited that the article was part of a larger project: “This study pertains to a larger research project aimed at exploring the effects of different RT (resistance training) intensities on blood biomarkers and muscular strength.” Furthermore, in the Methods and Discussion Sections (Juesas et al., 2023) we explained that we included only P1NP and BCTX/1000 (although we measured many more biomarkers variables) because these biomarkers are linked to bone health, which was an outcome of the first manuscript. We also cited in the present manuscript, that this study was part of a large project and cited the article of Juesas et al. (2023) (lines 118-122).

We understand our reviewer can doubt the ethical robustness (e.g., duplication and/or salami slicing) associated with publishing two manuscripts with similar samples as discussed by previous research (Altay & Koçak, 2021). Regarding this, we would like to highlight the actions (apart from those aforementioned) we have taken to ensure consistency and robustness, and avoid ethical concerns:  

As we mentioned in R1, although the same subjects were employed, the variables included in both manuscripts are different. The considerably high number of variables presented in both articles prevented us from designing only one manuscript since some journals limit the number of words and tables. Thus, we believe that the feasibility of publishing this large project in only one article is impossible. Moreover, we have not concealed at any time that the subjects and procedures to obtain the sample were the same. We cited in Figure 1 and Table 1 the aforementioned published manuscript (Juesas et al., 2023), followed by an explanation: “Participants are presented according to previous studies of this same project [29].”

These explanations added to Figure 1 and Table 1 were recommended by the Editor in their desk revision. They asked us to explain the similarities between both articles. After preparing a response letter, the Editor accepted our manuscript for peer review. Additionally, as suggested by editors, we limited the manuscripts’ similarities to match the journal’s acceptable text similarity. In the response letter, we indicated that, although the procedures are the same, the research questions are different, and the results may be applied and used by professionals from different areas. More specifically, the article by Juesas et al. (2023) is focused on functionality, strength, and parameters associated with frailty and could be, therefore, of interest to sports scientists, physiotherapists, and/or professionals in charge of the physical aspects of the elder population. On the other hand, this new article sent to Healthcare is more focused on physiological parameters (i.e., hepatic parameters, oxidative stress, inflammation) and, therefore, could be of interest to physicians or doctors to help in the management of conditions associated with physiological parameters. This is one of the reasons why we divided the outcomes into two manuscripts.  Therefore, we consider that this kind of combined study can reinforce and give light to deep-sea water supplementation, which is scarce when it is combined with resistance training.

That said, we sincerely hope that the Reviewer and Editors can consider our manuscript as an appropriate unit of publication in Healthcare. Thank you for your time.

Additional references cited:

Altay, S., & Koçak, Z. (2021). Multiple Publications From the Same Dataset: Is It Acceptable? Balkan Medical Journal, 38(5), 263–264. https://doi.org/10.5152/balkanmedj.2021.21008

Franzke, B., Schober-Halper, B., Hofmann, M., Oesen, S., Tosevska, A., Strasser, E. M., Marculescu, R., Wessner, B., & Wagner, K. H. (2019). Fat Soluble Vitamins in Institutionalized Elderly and the Effect of Exercise, Nutrition and Cognitive Training on Their Status-The Vienna Active Aging Study (VAAS): A Randomized Controlled Trial. Nutrients, 11(6), 1333. https://doi.org/10.3390/nu11061333

Franzke, B., Schober-Halper, B., Hofmann, M., Oesen, S., Tosevska, A., Henriksen, T., Poulsen, H. E., Strasser, E. M., Wessner, B., & Wagner, K. H. (2018). Age and the effect of exercise, nutrition and cognitive training on oxidative stress - The Vienna Active Aging Study (VAAS), a randomized controlled trial. Free radical biology & medicine, 121, 69–77. https://doi.org/10.1016/j.freeradbiomed.2018.04.565

Juesas, A., Gargallo, P., Gene-Morales, J., Babiloni-López, C., Saez-Berlanga, A., Jiménez-Martínez, P., Casaña, J., Benitez-Martinez, J. C., Ramirez-Campillo, R., Chulvi-Medrano, I., & Colado, J. C. (2023). Effects of Microfiltered Seawater Intake and Variable Resistance Training on Strength, Bone Health, Body Composition, and Quality of Life in Older Women: A 32-Week Randomized, Double-Blinded, Placebo-Controlled Trial. International journal of environmental research and public health, 20(6), 4700. https://doi.org/10.3390/ijerph20064700